# Flexible Capacitive Pressure Sensor Based on a Double-Sided Microstructure Porous Dielectric Layer

**DOI:** 10.3390/mi14010111

**Published:** 2022-12-30

**Authors:** Qingyang Yu, Jian Zhang

**Affiliations:** 1College of Control Science and Engineering, China University of Petroleum, Qingdao 266580, China; 2Morningcore Holding Co., Ltd., Qingdao 266400, China

**Keywords:** flexible capacitive pressure sensor, double-sided microstructure, hemisphere microstructure, porous structure, pressure sensing, object recognition

## Abstract

In the era of intelligent sensing, there is a huge demand for flexible pressure sensors. High sensitivity is the primary requirement for flexible pressure sensors, whereas pressure response range and resolution, which are also key parameters of sensors, are often ignored, resulting in limited applications of flexible pressure sensors. This paper reports a flexible capacitive pressure sensor based on a double-sided microstructure porous dielectric layer. First, a porous structure was developed in the polymer dielectric layer consisting of silicon rubber (SR)/NaCl/carbon black (CB) using the dissolution method, and then hemisphere microstructures were developed on both sides of the layer by adopting the template method. The synergistic effect of the hemispheric surface microstructure and porous internal structure improves the deformability of the dielectric layer, thus achieving high sensitivity (3.15 kPa^−1^), wide response range (0–200 kPa), and high resolution (i.e., the minimum pressure detected was 27 Pa). The proposed sensing unit and its array have been demonstrated to be effective in large-area pressure sensing and object recognition. The flexible capacitive pressure sensor developed in this paper is highly promising in applications of robot skin and intelligent prosthetic hands.

## 1. Introduction

In recent years, flexible pressure sensors based on different sensing mechanisms (piezoresistive, capacitive, piezoelectric, triboelectric, and magnetic) have been developed one after another owing to advances in flexible sensing [1,2,3,4,5]. Because of their good flexibility, high sensitivity, and low cost, flexible pressure sensors have been applied in flexible electronic skin, wearable health-monitoring devices, human motion state detection, prosthetic hands, human–computer interaction, etc. [6,7,8,9,10]. Compared with other sensors, flexible capacitive pressure sensors have received more attention due to their simple structure, high sensitivity, rapid dynamic response, and excellent stability [11,12,13]. Conventional capacitive flexible pressure sensors have sandwiched structures, and they mainly rely on the deformation of the middle dielectric layer to produce a rapid response to pressure, but because they are limited by the poor deformability of the elastic dielectric layer, such sensors still have a poor performance in terms of small-pressure detection and detection range [14,15]. To significantly improve both properties, advanced preparation processes and methods (e.g., the mold method and photolithography [16,17]) are often employed to form regular/irregular microstructure arrays (e.g., pyramids, semicircles, columns, and randomly distributed spine structures, folds [18,19,20,21,22]) on the surface of dielectric-layer elastomers (e.g., polydimethylsiloxane (PDMS), SR/Ecoflex) to improve deformability [23,24]. With this type of dielectric layer, the flat dielectric layer, which has poor compressibility, can produce a relatively large deformation in the range of small pressures, thus significantly improving the initial sensitivity of the sensor and increasing the resolution of the sensed pressure [25]. However, at large applied pressures, the deformation of the surface microstructure reaches its limit, and it is difficult for the sensor to realize a wide detection range. To improve the large deformation capability of the dielectric layer, a porous structured substrate (e.g., foam, sponge) is used [26,27,28]. This high porosity makes it easy for the sensor to deform under pressure and enables the pressure sensor to have a wide range of capabilities. However, to ensure large deformation under the applied pressure, this existing dielectric layer is often made too large compared to the electrode. The obvious disadvantage of relying on this approach is that it makes sensor packaging difficult, and the potential problem is that the service life is reduced due to structural instability over long-time use, thus affecting the long-term reliability of the sensor. Therefore, to achieve high sensitivity of the sensor while meeting the performance requirement of small-pressure detection and a wide detection range, from the perspective of improving the deformability of the dielectric layer, a double-sided microstructure combining the surface microstructure and internal microstructure is an effective solution.

This study reports a flexible capacitive pressure sensor based on a double-sided microstructure porous dielectric layer. To increase the dielectric constant of the dielectric layer, the dielectric layer was made of a mixture of silicon rubber (SR), salt (NaCl), and carbon black (CB) as a flexible substrate to create an internal porous microstructure by using the characteristics of salt to dissolve water easily. Meanwhile, the hemispherical microstructure was replicated by the template method under the top and bottom of the dielectric layer. Owing to the synergistic effect of the surface hemispheric microstructure and the internal porous structure, the sensor exhibits high sensitivity (3.15 kPa^−1^), high pressure resolution (i.e., the minimum pressure detected is 27 Pa), and a wide pressure detection range (0–200 kPa). Based on the proposed sensing unit, a large-area pressure sensing array is developed to map the pressure size and position distribution of the object. In addition, by adopting machine learning, object shape recognition is achieved with the dataset collected from the pressure array.

## 2. Materials and Methods

### 2.1. Materials

The silicon rubber (SR) and curing agent were purchased from Donghong Craft Material Co., Dongguan, China. The CB (model: BP2000) was purchased from Cabot Corporation (Boston, MA, USA). NaCl was purchased from Shandong Daiyue Salt Co. (Tai’an, China). The conductive silver paste with a volume resistance of 0.2 mΩ was purchased from Guangzhou Kaixiang Electronics Co. (Guangzhou, China). The polyethylene terephthalate film (thickness = 0.025 mm) was purchased from RuiXin Plastic Co. ( Jiaxing, China). The PET-based double-sided tape (model: 3M9495LE) and the polyimide single-sided tape (thickness = 50 µm) were purchased online. All materials were used without additional treatment.

### 2.2. Preparation

#### 2.2.1. Microstructure Mold

An array hemispheric elastic microstructure mold and a surface microstructure-free mold of the same size were fabricated by 3D printing. Specifically, the diameter of hemispheres and the spacing between them were 1.5 mm, the microstructure area was 11 × 11 cm, and the whole mold size was 14 × 14 cm. The microstructure of the mold is shown in Figure 1.

#### 2.2.2. Composite Material

The mass ratio of SR: NaCl: CB: SR curing agent was set to 50:25:1:1. First, 60 g of SR was added to the beaker, and then 30 g of NaCl powder was added to the beaker with mechanical stirring for 1 h. Then, 1.2 g of CB was added to the beaker and stirred for 40 min. Finally, 1.2 g of curing agent was added to the beaker and stirred for 20 min to mix all the ingredients. The beaker was then placed in a vacuum-drying oven for 10 min to eliminate air bubbles. 

### 2.3. Fabrication of the Dielectric Layer

The fabrication process of the porous double-sided microstructure dielectric layer is shown in Figure 2a. First, the composite material was poured on the surface of mold A with the microstructure adsorbed on the spin coater tray, and then it was spun at 600 rpm for 40 s. Next, the finished mold was placed in a vacuum-drying oven to remove the air (10 min). The above steps were repeated to cover the exposed part of the mold with a uniform layer of composite material, and the spin-coating-vacuuming process was repeated until the entire surface of the microstructure mold was covered with a uniform layer of composite material after vacuuming. This operation of mold A was repeated for mold B. Then, the sides of mold A and mold B containing the composite material (not fully cured yet) were aligned and attached, and the molds were covered with an object of 1 kg that was slightly larger than the molds (to make the two molds fit more tightly). Then, the molds were placed in a vacuum-drying oven for 10 min to remove the air between them to obtain a double-sided microstructure porous dielectric layer. The fabrication process of the porous single-sided microstructure dielectric layer is shown in Figure 2b. The operation of mold A was repeated for mold C to make a single-sided microstructure dielectric layer. Mold D without the microstructure was used to manufacture the porous dielectric layer without the microstructure, as shown in Figure 2c. Afterward, all molds were transferred to an oven for thermal curing (50 °C, 4 h). Subsequently, all the molds were placed in an ultrasonic cleaner with deionized water for 2 h to separate the dielectric layer from the molds. After stripping, all dielectric layers were further shaken for 24 h to remove the internal NaCl. Eventually, the porous double-sided microstructure dielectric layer, the porous single-sided microstructure dielectric layer, and the porous dielectric layer without the microstructure were obtained.

### 2.4. Fabrication of the Electrode Layer

The electrode layer was prepared by the screen-printing method, and the fabrication process is illustrated in Figure 3a. The flexible substrate PET and the printing screen were fixed on the screen-printing platform, and the conductive silver paste was printed onto the entire electrode pattern surface using a scraper. This process was repeated several times to ensure that the complete electrode pattern was formed on the PET surface, and then the printed PET was placed in a drying oven to cure at 80 °C for 8 min. The actual electrode layer is shown in Figure 3b. The electrode area was a 10 ×10 mm square, and a 6 × 10 mm wire area was led out from one side of the square electrode to facilitate the lead-out for subsequent sensor performance testing. 

### 2.5. Sensor Packaging

Figure 4a shows the packaging process of the sensor. The porous double-sided microstructure dielectric layer, the porous single-sided microstructure dielectric layer, and the porous dielectric layer without the microstructure were cut into appropriate sizes and sandwiched between the two printed silver electrodes, respectively. In addition, the PI tape was used to encapsulate the sensor. The conductive aluminum tape connects the conductor area reserved for the electrode to form a pressure sensor. These sensors are labeled as sensor I, sensor II, and sensor III, and their entities are presented in Figure 4b.

### 2.6. Fabrication of the Sensor Pressure Array

Following the process of sensor unit fabrication, the electrode array pattern shown in Figure 5a was designed and processed into a printing screen. Then, the conductive silver paste was scraped and cured with heat to obtain the actual electrode array. Meanwhile, conductive silver glue was used to connect the array electrode to the terminal wire. The steps for making the dielectric layer are the same as those for the dielectric layer of sensor Ⅰ. Afterward, the upper array electrode, middle dielectric layer, and lower array electrode were assembled to enhance the insulation between the upper and lower array electrodes and to improve the performance of the flexible capacitive pressure sensor array. The non-conductive side of the pattern 2 array electrode was fitted together with the dielectric layer, and the conductive side was coated with TPU to make it insulated from the outside world. Finally, the sensor pressure array was encapsulated using 3M tape, and its entirety is shown in Figure 5b.

### 2.7. Characterization and Measurements

The hemisphere structure of the dielectric layer surface and the surface morphology of the internal porous microstructure were characterized using FESEM. Meanwhile, the capacitance variation was measured at 1 kHz with an LCR meter (TH2826). The external dynamic pressure was applied using a computer-controlled tensile tester (ZQ-990B) (mainly to test the repeatability and response time). In addition, static pressure (test sensitivity) was applied using a manual digital push–pull gauge (HP-50).

## 3. Results and Discussions

### 3.1. Finite Element Analysis of Dielectric Layer Sensitivity

The Workbench Platform (ANSYS) was used for the statics simulation of sensor I, sensor II, and sensor III to investigate the reason why the porous double-sided microstructure dielectric layer was most effective for sensitivity enhancement. The three-dimensional model of the sensor created by SolidWorks was imported into the Workbench software. The density of the dielectric layer was set to 0.98 g/cm^3^, Young’s modulus was 1.2 GPa, and Poisson’s ratio was 0.4. During the simulation, the bottom electrode of the sensor was fixed, and the static force was applied to the top electrode surface, ignoring the deformation of the electrode layer. The mechanical simulation results of the sensor at a pressure of 500 Pa, 1 kPa, 5 kPa, and 10 kPa are shown in Figure 6 (the top electrode is hidden). It can be seen that under the same pressure conditions, the deformation of the sensor dielectric layer is ranked as sensor I > sensor II > sensor III. The reason for this difference is that the force on the dielectric layer of sensor I is mainly concentrated at the array elastic hemispheric microstructure on the upper surface and the array elastic hemispheric microstructure on the lower surface; the force on the dielectric layer of sensor II is mainly concentrated at the array elastic hemispheric microstructure on the upper surface and the whole lower surface; the force on the dielectric layer of sensor III is mainly concentrated at the whole upper surface and the whole lower surface. Meanwhile, the force area of the sensor dielectric layer is ranked as sensor I > sensor II > sensor III. In the case of the same force, the smaller the force area, the greater the pressure on the object. Thus, the dielectric layer of sensor A generates the largest deformation, the dielectric layer of sensor B generates the second-largest deformation, and the dielectric layer of sensor C generates the smallest deformation. From the above simulations, the sensor sensitivity is ranked as sensor I > sensor II > sensor III. Theoretically, it is proved that the porous double-sided microstructure dielectric layer has the best effect on improving the sensitivity of the flexible capacitive pressure sensor.

### 3.2. Sensitivity

A digital push–pull gauge was used to load pressure on the sensor surface, and an LCR meter was employed to collect the corresponding output capacitance. Figure 7 shows the pressure–capacitance curves of the sensor. The sensitivity (*S*) of the capacitance sensor can be defined as [29]:*S* = δ(ΔC/C_0_)/δP(1)
where ΔC is the variation of capacitance (ΔC = C − C_0_), C is the capacitance under the corresponding pressure, C_0_ is the initial capacitance, and P is the applied pressure. According to Equation (1), the sensitivity of the sensor is the slope of the curve. Then, the sensitivities of the above sensors in different pressure intervals were obtained by the function of segmented linear fitting, and the results are shown in Figure 8. It can be seen that the sensitivity of sensor I is the highest in any pressure range. The experimental results indicate that the porous double-sided microstructure dielectric layer has the best effect in terms of sensitivity.

### 3.3. Characterization

Figure 9 shows the SEM image of the dielectric layer cross-section of sensor Ⅰ. It can be observed that there are many holes of different sizes in the interior of the dielectric layer. According to the local magnification, the pore diameter ranges from 10 to 160 μm. Such internal holes greatly improve the elastic performance, which leads to a larger deformation at the same pressure compared to the dielectric layer without holes, thus increasing the sensitivity of the flexible capacitive pressure sensor. In addition, the combination of the porous microstructure inside the dielectric layer and the array elastic microstructures on the surface of the dielectric layer greatly improves the sensitivity of the flexible capacitive pressure sensor. Note that the array elastic hemispheric microstructure (yellow) on the surface of the dielectric layer is not a true hemisphere, and this is caused by the fact that the 3D printing technology produces the mold by printing in layers. As a result, the array elastic hemispheric microstructure mold made is not a true hemisphere, which makes the array elastic microstructure on the surface of the dielectric layer approximate a hemisphere. Overall, among the three types of dielectric layers, the porous double-sided microstructure dielectric layer achieves the best effect in improving the sensitivity of the flexible capacitive pressure sensor, and the sensitivity of the sensor has been greatly improved, which proves that the method is effective in improving the sensitivity of the flexible capacitive pressure sensor.

### 3.4. Dynamic Performance

The pressure response curves are shown in Figure 10a,b, with the square wave pressure and the sharp wave pressure of 0.9 kPa, 6 kPa, 38 kPa, and 52 kPa applied to the sensor surface. It can be seen that when a steady-state force is applied to the sensor, the capacitance change remains stable during the application of the pressure, and there is a clear distinction in the capacitance change for different steady-state forces. When a transient force is applied to the sensor, the capacitance change is almost the same as that of applying a steady-state force of the same size. The above results indicate that the sensor has a stable response to different loading types of pressure and meets the requirement for stability. In addition, we tested the minimum pressure limit by successively loading different weights (100 mg, 200 mg, 500 mg) onto the surface of the sensor. When a 500 mg weight was loaded, the sensor had a significant capacitance output change, as shown in Figure 10c. At this time, the minimum pressure that the sensor can respond to is about 27 Pa. It can be observed that the sensor has a high resolution, which can reflect the small pressure variations well. The sensor pressure response curves for the applied/released 0.9 kPa are shown in Figure 10d–f. The results show that the sensor has a short response time and a recovery time of 120 ms, so it meets the requirement for rapid response to pressure. The sensor resolution refers to the minimum pressure that the sensor can respond to. To test the service life and durability of the sensor, a continuous pressure of 1.5 kPa was applied/released in a time period of 2500 s, and the results are shown in Figure 10g–i. It can be seen that the overall graphs of the sensor’s capacitance variation curves are stable without large fluctuations, indicating that the sensor has a reliable service life and durability. Meanwhile, the curve graphs of the local repeatability tests are shown for 220–230 s and 1960–1970 s, respectively. The curves of the sensor capacitance change at the beginning and the end of the applied pressure are almost the same, which is good proof of its reliable service life and durability. The above performance tests indicate that the porous double-sided microstructure dielectric layer enables the flexible capacitive pressure sensor to achieve high sensitivity and excellent performance in terms of detection range and resolution.

### 3.5. Application

To verify the ability of the flexible capacitive pressure sensor array to sense the real object, two 10 g weights, hexagonal bars, and discs in the form of pressure point, pressure line, and pressure plane were placed on its surface for testing. The results are illustrated in Figure 11. It can be seen that the flexible capacitive pressure sensor array can reflect both the size of the object pressure and the location distribution of the pressure on the surface of the object, with varying response levels and response areas for different objects.

Object shape recognition plays an important role in robotic hand-grasping perception tasks [30,31]. There are two main methods of object recognition at present: One is an object recognition system based on visual perception, whose performance mainly depends on the quality of the image and the quality of the recognition algorithm. Another method is an object recognition system based on tactile perception. This method is not easily affected by a complex environment, and its direct contact with objects makes the perceived information more abundant and reliable. The flexible sensor array is flexible and easy to integrate. When grasping an irregular object, the flexible sensor array can bend and fold freely to fully contact the irregular object, and the perceived data is more abundant and reliable. Therefore, this study prepared a data glove by integrating a flexible capacitive pressure sensor array into the fingertips of each of the five fingers and the palm of the glove, as shown in Figure 12a. By combining machine learning and pressure sensing to achieve object recognition, the process of the human hand’s tactile perception of external information was simulated. The object recognition system was built based on the integration of the data glove, the data collection circuit, and the object recognition model, as shown in Figure 12b. The working principle is as follows: the control chip (model: TMS320F28388) controls the single knife double-throw switch (model: ADG711) to select the pressure sensor array unit one by one; meanwhile, it controls the capacitive signal collection chip (model: PCap02) to collect the signal by the flexible capacitive pressure sensor array and transmits the signal to the host computer through the serial port. The data is preprocessed in Matlab, and the object recognition model is established using a probabilistic neural network (PNN) algorithm. The feature data of six objects (baseball (No. 1), orange (No. 2), mineral water (No. 3), anhydrous ethanol bottle (No. 4), pen (No. 5), and hexagon bar (No. 6)) were collected from the time of grasping and stabilizing to the time of starting to release the objects, as shown in Figure 13. In this approach, a total of 200 sets of data were collected for each object, and a total of 1200 sets of data were collected, each of which contained 94-dimensional features. Then, 80% of the data was randomly selected as the training set of the PNN algorithm to train the object recognition model, and the remaining 20% of the data was used as the test set to test the recognition accuracy of the object recognition model.

The overall structure of the PNN algorithm is shown in Figure 14a. The input layer of the model has 71 nodes, which can input 71 eigenvalues. The hidden layer contains 960 neurons. The output layer outputs 1–6, representing six different objects, respectively. The test results are presented in Figure 14b, where “*” represents the real class of the object, and “△” represents the classification result of the object recognition model. It can be seen that Object 1 and Object 2, Object 3 and Object 4, and Object 5 and Object 6 have more recognition errors than other cases. To see more clearly the classification of objects with the object recognition model, the confusion matrix was drawn, as shown in Figure 14c. As shown in the figure, since Object 1 and Object 2, Object 3 and Object 4, and Object 5 and Object 6 are similar in shape, they are easily confused with each other; meanwhile, the probability of confusion between other objects is very low. Table 1 lists the tested recognition accuracy of the object recognition model. It can be seen that the recognition accuracy of all objects exceeds 84%, and the overall recognition accuracy is 87.1%. The results indicate that based on the proposed flexible capacitive pressure sensor array, an adequate and accurate data set is collected to guarantee a high recognition accuracy of the object recognition system, which is expected to provide an accurate tactile perception of the prosthetic hand with the assistance of machine learning.

## 4. Conclusions

In this work, a flexible capacitive pressure sensor based on a double-sided microstructured porous dielectric layer was developed. Owing to the synergistic effect of the surface hemispherical microstructure and the internal porous structure, the sensor has high sensitivity (3.15 kPa^−1^), a wide pressure sensing range (0–200 kPa), and high resolution (i.e., the minimum pressure detected is 27 Pa). Meanwhile, the array sensor satisfies the three force-loading forms of pressure perception: “point”, “line”, and “surface”. The sensor array is integrated into the glove to obtain the pressure dataset of the grasped object, and it is combined with machine learning to recognize the shape of the grasped object, achieving an overall accuracy is 87.1%. These results indicate that the sensor has the potential for intelligent prosthetic hand haptic perception. Overall, a reasonable combination of multiple microstructures provides an effective approach for improving sensor performance.

## Figures and Tables

**Figure 1 micromachines-14-00111-f001:**
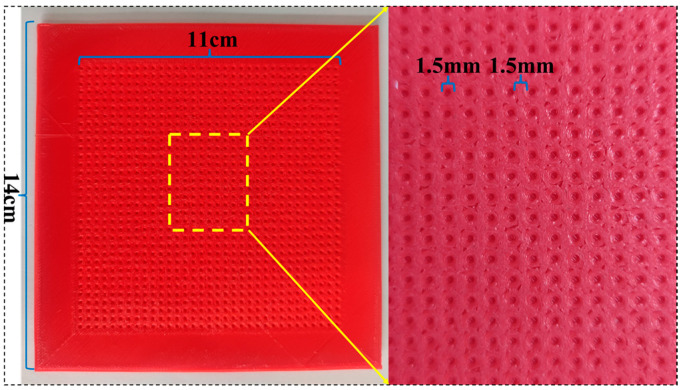
Actual image of microstructure mold.

**Figure 2 micromachines-14-00111-f002:**
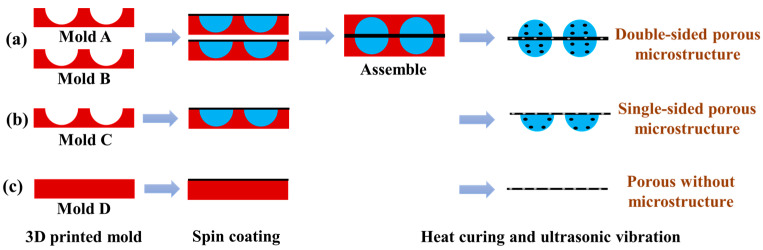
Fabrication process of dielectric layer. (**a**) The porous double-sided microstructure dielectric layer; (**b**) the porous single-sided microstructure dielectric layer; (**c**) the porous dielectric layer without microstructure.

**Figure 3 micromachines-14-00111-f003:**
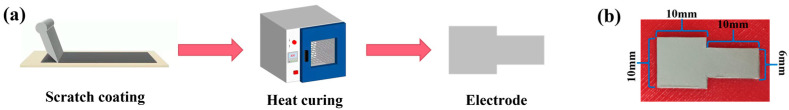
Fabrication of the electrode layer. (**a**) The fabrication process; (**b**) the actual image.

**Figure 4 micromachines-14-00111-f004:**
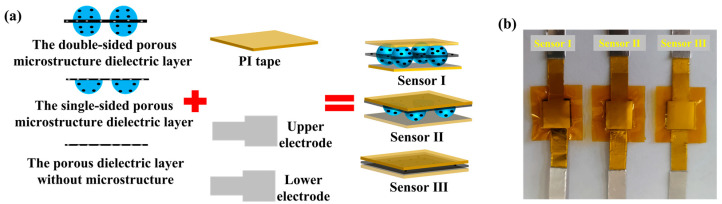
Sensor packaging of the sensor I, the sensor II and the sensor III. (**a**) The packaging process; (**b**) the actual image.

**Figure 5 micromachines-14-00111-f005:**
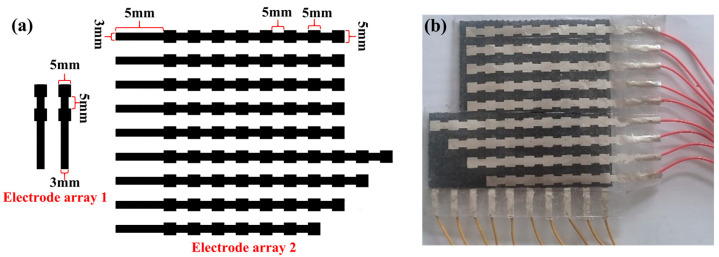
Fabrication of the sensor pressure array. (**a**) The electrode array pattern; (**b**) the actual image.

**Figure 6 micromachines-14-00111-f006:**
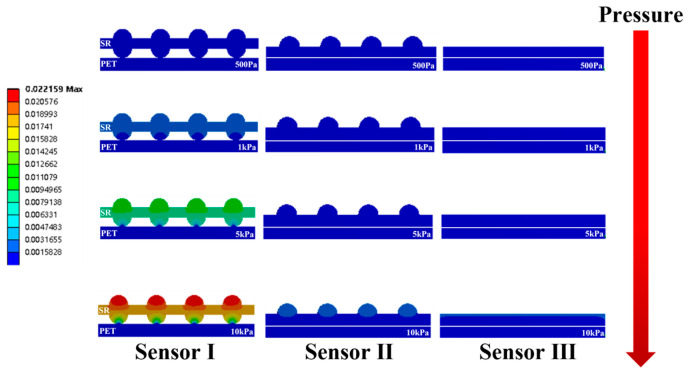
Finite element diagram of sensor I, sensor II, and sensor III under pressures of 0.5 Pa, 1 kPa, 5 kPa, and 10 kPa.

**Figure 7 micromachines-14-00111-f007:**
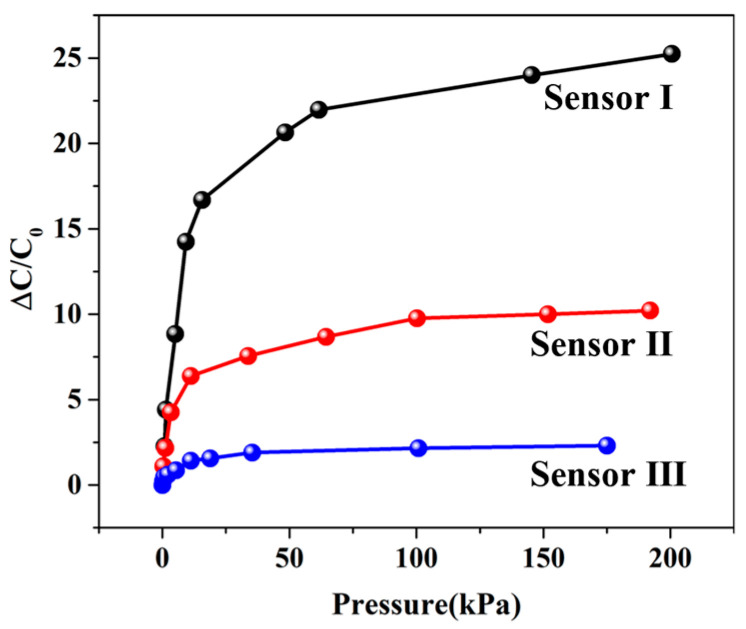
Pressure response curve of the sensor.

**Figure 8 micromachines-14-00111-f008:**
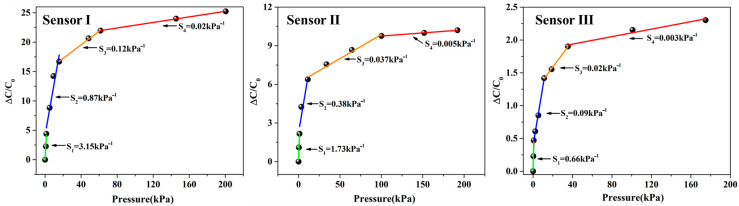
The function of segmented linear fitting.

**Figure 9 micromachines-14-00111-f009:**
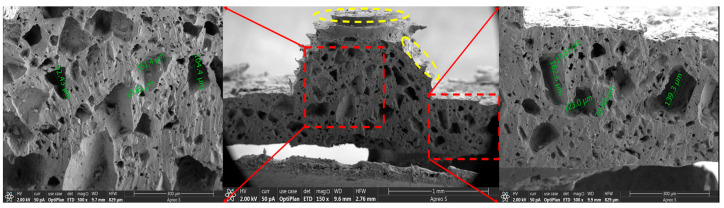
Cross-section SEM image of the dielectric layer of sensor Ⅰ (the porous microstructure in the red area, and the hemispherical microstructure in the yellow area).

**Figure 10 micromachines-14-00111-f010:**
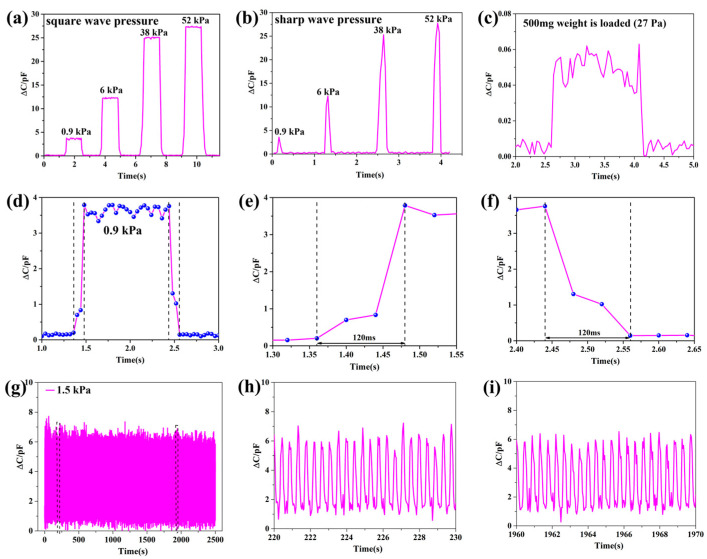
Dynamic performance of the capacitive pressure sensors. (**a**) The output curve under the square wave pressure of 0.9 kPa, 6 kPa, 38 kPa, and 52 kPa; (**b**) the output curve under the sharp wave pressure of 0.9 kPa, 6 kPa, 38 kPa, and 52 kPa; (**c**) minimum response curve of force for 27 Pa; (**d**) the sensor force response curves for 0.9 kPa; (**e**) response time of the sensor; (**f**) recovery time of the sensor; (**g**) repeatability curve of the sensor under the pressure of 1.5 kPa; (**h**) local repeatability curve within 220–230 s; (**i**) local repeatability curve within 1960–1970 s.

**Figure 11 micromachines-14-00111-f011:**
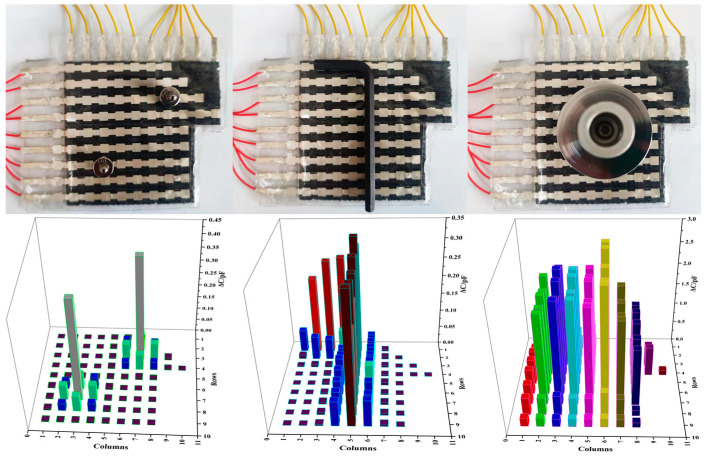
Shape perception of different objects by pressure sensor array.

**Figure 12 micromachines-14-00111-f012:**
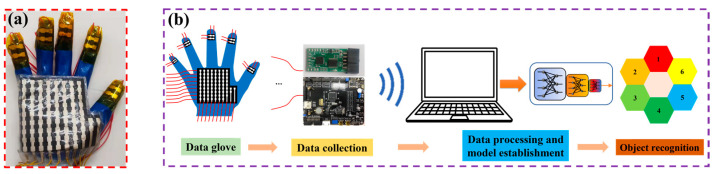
The process of object recognition. (**a**) The data glove with integrated sensors; (**b**) the object recognition system.

**Figure 13 micromachines-14-00111-f013:**
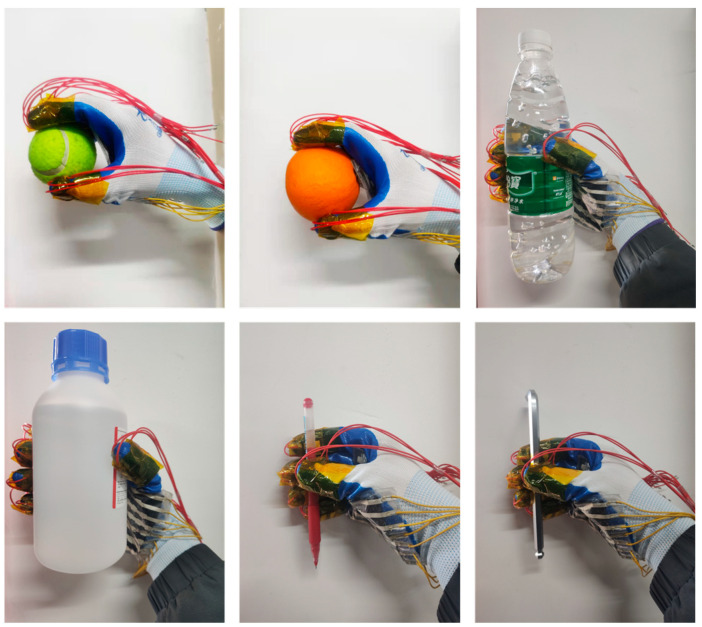
The process of grasping six objects (baseball, orange, mineral water, anhydrous ethanol bottle, pen, and hexagon bar).

**Figure 14 micromachines-14-00111-f014:**
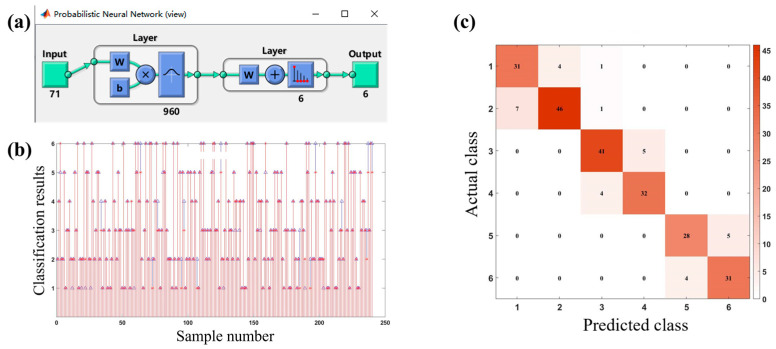
Building and testing of object recognition model. (**a**) Overall structure of PNN algorithm; (**b**) test results of sample data; (**c**) confusion matrix.

**Table 1 micromachines-14-00111-t001:** Recognition accuracy of object recognition model.

Object Number	Recognition Accuracy	Overall Recognition Accuracy
1	86.1%	87.1%
2	85.2%	
3	89.1%	
4	88.9%	
5	84.8%	
6	88.6%	

## Data Availability

Not applicable.

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
