# Peer review of "Flexible Capacitive Pressure Sensor Based on a Double-Sided Microstructure Porous Dielectric Layer"

_micromachines, 2022, doi:10.3390/mi14010111_

Round 1

Reviewer 1 Report

Refer to file for comments.

Reviewer 2 Report

In the present manuscript, authors have have  fabricated flexible capacitive pressure sensor arrays and device performance was measured. Manuscript is well written and provide sufficient data with quality. Reviewer has positive opinion for acceptance of the manuscript. However, I have one concern regarding device endurance. Can they show active layer and electrode microstructural features after sufficient number of flexibility test of the device? However, this is not a strict requirement.  

Round 2

Reviewer 1 Report

The SEM figure has some dimensions/text that is difficult to read. If it's not important to the narrative, consider using SEM images without the measurements.

Overall the improvements are good.